# Comparison of Three Mixed-Effects Models for Mass Movement Susceptibility Mapping Based on Incomplete Inventory in China

Yifei He [1,2] and Yaonan Zhang [1,2,3,*]

1 Northwest Institute of Eco-Environment and Resources, CAS, Lanzhou 730000, China
2 National Cryosphere Desert Data Center, Lanzhou 730000, China
3 Gansu Data Engineering and Technology Research Center for Resources and Environment, Lanzhou 730000, China
* Correspondence: yaonan@lzb.ac.cn

**Abstract:** Generating an unbiased inventory of mass movements is challenging, particularly in a large region such as China. However, due to the enormous threat to human life and property caused by the increasing number of mass movements, it is imperative to develop a reliable nationwide mass movement susceptibility model to identify mass movement-prone regions and formulate appropriate disaster prevention strategies. In recent years, the mixed-effects models have shown their unique advantages in dealing with the biased mass movement inventory, yet there are no relevant studies to compare different mixed-effects models. This research compared three mixed-effects models to explore the most plausible and robust susceptibility mapping model, considering the inherently heterogeneously complete mass movement information. Based on a preliminary data analysis, eight critical factors influencing mass movements were selected as basis predictors: the slope, aspect, profile curvature, plan curvature, road density, river density, soil moisture, and lithology. Two additional factors, namely, the land use and geological environment division, representing the inventory bias were selected as random intercepts. Subsequently, three mixed-effects models—Statistical-based generalized linear mixed-effects model (GLMM), generalized additive mixed-effects model (GAMM), and machine learning-based tree-boosted mixed-effects model (TBMM)—were adopted. These models were used to evaluate the susceptibility of three distinct types of mass movements (i.e., 28,814 debris flows, 54,586 rockfalls and 108,432 landslides), respectively. The results were compared both from quantitative and qualitative perspectives. The results showed that TBMM performed best in all three cases with AUROCs (Area Under the Receiver Operating Characteristic curve) of cross-validation, spatial cross-validation, and predictions on simulated highly biased inventory, all exceeding 0.8. In addition, the spatial prediction patterns of TBMM were more in line with the natural geomorphological underlying process, indicating that TBMM can better reduce the impact of inventory bias than GLMM and GAMM. Finally, factor contribution analysis showed the key role of topographic factors in predicting the occurrence of mass movements, followed by road density and soil moisture. This study contributes to assessing China's overall mass movement susceptibility situation and assisting policymakers in master planning for risk mitigation. Further, it demonstrates the tremendous potential of TBMM for mass movement susceptibility assessment, despite inherent biases in the inventory.

**Keywords:** nationwide; susceptibility mapping; mass movement; inventory bias; tree-boosted; mixed-effects models

## 1. Introduction

China is the country that experiences the most frequent natural disasters globally, as it witnesses numerous hazards such as floods, droughts, earthquakes, sandstorms and wildfires every year [1,2]. The landscape of China is characterized by widespread

mountainous areas, which makes it highly prone to mass movements [3]. Mass movements such as debris flows, rockfalls and landslides are major geological disasters that have devastating effects on property, human life and the country's ecological environment [4,5]. It has been recognized that mass movements may pose an even more serious threat in the future due to the potential effects of rapid urbanization, compounded by aggravated climate change [6]. Therefore, a credible nationwide mass movement susceptibility map is paramount for providing a generalized overview of potential mass movement propagation areas in China.

Over the past few decades, an increasing number of studies on mass movement susceptibility assessments have been published, most of which have been performed on local areas [7–9]. However, in order to improve the overall perception of mass movement risk, some studies have begun to assess susceptibility in very large areas, including national-scale evaluations [10–15], continental-scale analyses [16–19], and global-scale assessments [20–22]. It can be found that such evaluations for large areas are mostly based on statistical or machine learning models, as the lack of detailed geotechnical data limits the application of physical-based models [23]. Both statistical models and machine learning models assume that the conditions that caused mass movements in the past may lead to future mass movements; thus, the correlations between controlling factors and mass movement inventories of past events were fitted by models to determine the probability of future mass movement occurrence [24]. So far, many statistical models have been successfully applied to mass movement susceptibility mapping, such as weight of evidence [25], frequency ratio [26], logistic regression [27], information value [28] and generalized additive model [29]. Recently, many machine learning models have demonstrated excellent performance, such as support vector machine [30], artificial neural network [31], maximum entropy model [32,33], naïve Bayes [34], decision tree [35], random forest [36] and gradient boosted trees [37]. Each of these approaches has its own pros and cons. For example, as black-box models, random forest and gradient-boosted trees tend to show better performance but low interpretability.

During mass movement susceptibility modeling, many factors will affect the final assessment outcome, including the quality of the mass movement inventory [12,38,39], the selection of spatial mapping units and their resolutions [40], the sampling strategy for mass movement-free units [41], the selection of conditional factors and their quality [42], the choice of the susceptibility algorithm [8,43], the optimization of model parameters [44] and the model validation metrics [45]. Among them, a representative mass movement inventory is the key prerequisite to getting a reliable susceptibility map [46–49]. However, the available mass movement databases are often biased and incomplete. In general, the reported mass movements tend to be more representative in economically advanced areas with a large population [4,50]. Such areas have more abundant detection methods with programs for detailed investigations of mass movements. Thus, mass movements in densely populated or trafficked areas are more likely to be observed and reported. Suppose the data on mass movement is derived from the interpretation of optical remote sensing or LiDAR. In that case, it is typically overrepresented within forested areas and underestimated in regions with intense human activity, such as the presence of cultivated land [51]. Because of the distinctive morphological characteristics of mass movements in forest areas, they are easily identified. On the other hand, for the mass movements occurring within arable land, their topographical features are easily blurred or eliminated or altered by human activities. Thus, for a study region as large as China, with its complex topographic conditions and unbalanced population and economic development [52], the available mass movement inventory is usually heterogeneously complete.

The spatially heterogeneous completeness of mass movement information has an enormous impact on susceptibility assessment. Several studies have shown that if inventory bias is directly ignored, even models that perform well on quantitative metrics such as accuracy, F1-score, or AUROC will propagate this bias into the model results, leading to geomorphological implausibility in final spatial prediction patterns [48,53]. For example, Steger et al. [54] simulated a highly incomplete landslide inventory in forested areas. They

found that when simply ignored this bias, the models would predict landslide susceptibility to be low in forested areas, as opposed to predictions when the inventory was complete.

Some researchers have proposed using mixed-effects models, considered beneficial where measurements are repeated or are statistically relevant. Popular in medicine, ecology, and economy because of their unique advantages in analyzing hierarchical or longitudinal data [55,56], they are also helpful in mass movement modelling studies. In a study by Steger et al. [54], mixed-effects models successfully reduced the effects of inventory bias in mass movement susceptibility mapping. Unlike the traditional mass movement susceptibility models that use only fixed effects to predict susceptibility, mixed-effects models also incorporate random effects components. Introducing these additional components can account for spatially heterogeneous completeness of mass movement inventory, thereby counterbalancing the associated propagation of biases in the data. In addition, the fixed-effects term of the mixed effects model can also be implemented in many ways to achieve the desired results. The generalized linear mixed-effects model (GLMM) [53,54], based on the GLM, and the generalized additive mixed-effects model (GAMM) [57,58], based on the GAM, are examples of statistical-based mixed models that have performed well in this regard. Recently, Sigrist [59] has innovatively developed Tree-boosted mixed-effects model (TBMM) and demonstrated its excellent performance. However, no relevant research articles illustrate the suitability of this machine learning-based mixed model in the field of mass movement susceptibility assessment and its comparison with statistical-based mixed models.

Against the backdrop of the above discussions, this paper has used three mixed-effects models (i.e., statistical-based GLMM, GAMM and machine learning-based TBMM) to assess the mass movement susceptibility in selected areas of China. This study addresses how to account for the inherent spatial incompleteness of the inventory and compares the performance of the models, both from quantitative and qualitative perspectives and explores which of them show superior performance. The comparison is based on quantitative analysis of AUROCs of cross-validation, spatial cross-validation, and predictions on simulated highly biased inventory, as well as qualitative perspectives using spatial patterns of susceptibility maps.

## 2. Study Area and Materials

### 2.1. Characterization of the Study Area

Located in the eastern part of Eurasia along the Pacific Ocean coastline, China is the third largest country in the world, occupying a territory of about 9.6 million km². Its geographic coverage is approximately 73°–135°E longitude and 18°–54°N latitude (Figure 1). There are widely varying landscapes and climate zones in China. The mountainous areas (including mountains, hills and rugged plateaus) are vast and cover about 70% of the country's land area [3]. The terrain is generally characterized by a high in the west and a low in the east, with a "staircase"-type distribution consisting of three steps. The first step of the staircase is the Qinghai–Tibet Plateau (Tibet region), with an average height of over 4000 m. The Kunlun, Qilian and Hengduan Mountains are located on its northern and eastern edges, which mark the boundary between the first and second steps of the terrain. The second step (including the NW, Loess and SW regions) has huge basins and plateaus with an average elevation of 1000–2000 m. The Greater Khingan, Taihang, and Xuefeng Mountains are located to the east, forming a dividing line between the second and third steps of the terrain. The third step (including NE, Yangtze and SC regions) is dominated by vast plains and hills, most of which occur at elevations less than 500 m above sea level.

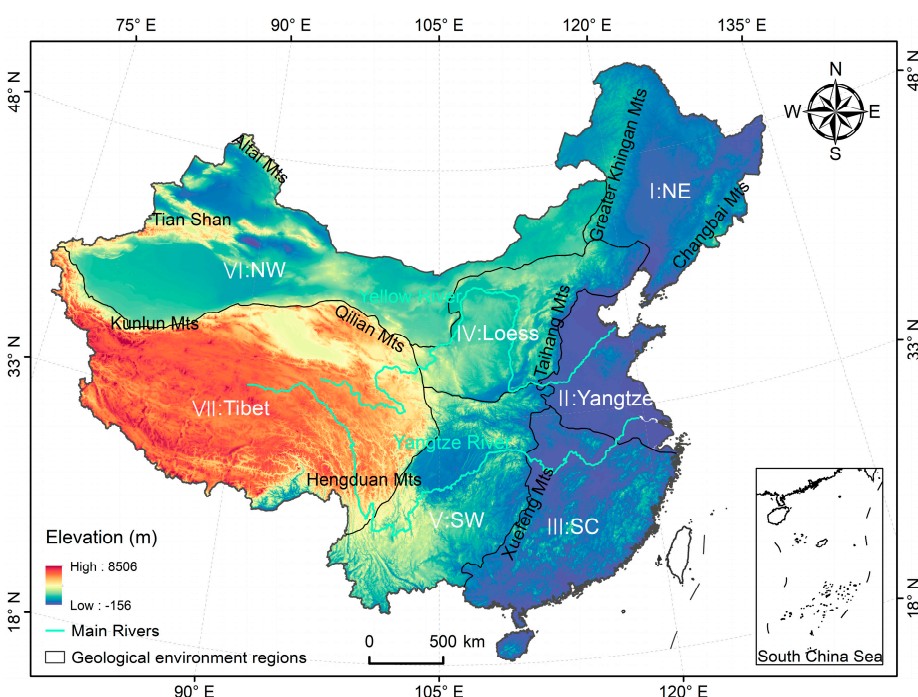

**Figure 1.** Overview map of China. The seven geological environment regions are shown as the Northeastern plain and mountain region (I: NE); the Huang–Huai–Hai–Yangtze River Delta plain region (II:Yangtze); the South China low mountain and hill region (III:SC); the North China Loess Plateau region (IV:Loess); the Southwest karst mountain region (V:SW); the Northwest mountain and basin region (VI:NW); and the Qinhai–Tibet Plateau region (VII:Tibet).

Due to the vastness of its territory and complex terrain characteristics, China exhibits variable climatic conditions. Its eastern part is significantly affected by the monsoon. This region exhibits tropical and subtropical climates (SW and SC region) and temperate monsoon climates (NE region) distributed from south to north. The western part is located inland, mainly with mountainous highland climate (Tibet region) and temperate continental climate (NW region) [60]. Precipitation generally declines from southeast to northwest in China. As for earthquakes, China lies between two of the world's most extensive seismic belts, i.e., the Circum-Pacific belt and the Alpide belt, with intense seismic activity [61]. Due to the widespread mountainous areas, rugged terrain, active seismicity and intense monsoons, China is prone to different geological disasters. Numerous landslides, rockfalls, and debris flows that occur here cause immeasurable casualties and economic damage [4].

### 2.2. Spatial Database

### 2.2.1. Inventory of Mass Movement

A spatial dataset representing former mass movements is essential for carrying out mass movement susceptibility mapping and hazard assessment [62]. Since 2005, the China Geological Survey has been conducting detailed investigations on six types of geological disasters, namely, landslide, rockfall, debris flow, ground subsidence, ground collapse and ground fissure. The survey consists of three main procedures: (i) interpretation of high-resolution remote sensing imagery; (ii) field verification; (iii) collating and correcting of obtaining data. The mass movement catalog used in this paper is credited to the National Geological Hazard Detailed Survey (https://geocloud.cgs.gov.cn/#/home accessed on 2 April 2022). In this work, the three most widely distributed and profound mass movements, including landslide (all slide-type movements), rockfall (all fall-type movements) and debris flow (all flow-type movements) were selected for analysis. There is a total of 108,432, 54,586 and 28,814 reported landslides, rockfalls and debris flows points, respectively. The inventory maps and kernel density maps for the three mass movements

are shown in Figure 2. It was apparent that the completeness of the mass movement in different regions is heterogeneous. For example, the very low mass movement density in the northwest China may be related to sparse populations and tough surveys.

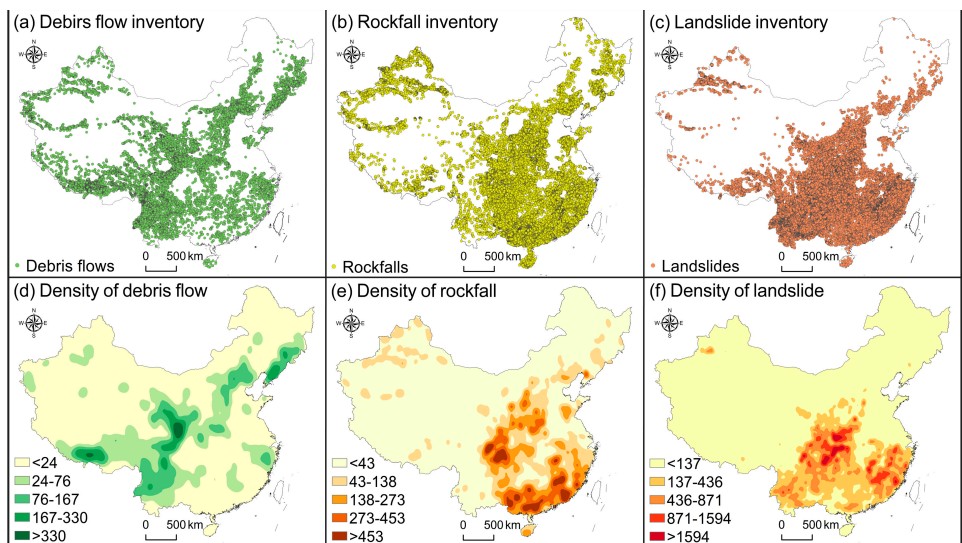

**Figure 2.** Spatial distribution maps and kernel density maps of three types of mass movements.

### 2.2.2. Mass Movement Influencing Factors

Mass movements' occurrence mechanism is extremely intricate and is affected by various conditional factors [7,63]. In the light of the relevant literature, six common categories of influencing factors, including topography (slope, aspect, profile curvature, plan curvature), human activities (road density), hydrology (river density, soil moisture), geology (lithology), land use and geological environment division are chosen as primary factors [4,10,57]. It should be emphasized that the mass movement data used in this paper are primarily triggered by heavy rainfall events, floods, earthquakes, or a combination thereof. Thus, considering the diversity of mass movement triggering conditions, we finally chose the common influencing factors of these mass movement as evaluation indicators [64,65]. The information of types and sources for mass movement inventory and influencing factors are shown in Table 1. Among these factors, slope, aspect, profile curvature, plan curvature, road density, river density, soil moisture and lithology are considered basic predictors (fixed-effect factors). In contrast, land use and geological environment division are considered as factors that are linked to the incompleteness of mass movement data (random intercept factors). The relationships between mass movement occurrence and conditional factors are described below. The fixed-effect and random intercept factors are also discussed in some detail.

**Table 1.** Data type and sources of mass movements inventory and influencing factors.

| Data | Original Data Type | Data Sources |
|---|---|---|
| Mass movements inventory | Point | China geological survey |
| Slope | Grid (90 m) | Derived from DEM https://srtm.csi.cgiar.org (accessed on 5 April 2022) |
| Aspect | Grid (90 m) | Derived from DEM |
| Profile curvature | Grid (90 m) | Derived from DEM |
| Plan curvature | Grid (90 m) | Derived from DEM |
| Road density | Line | https://www.tianditu.gov.cn (accessed on 9 April 2022) |
| River density | Line | https://www.tianditu.gov.cn (accessed on 9 April 2022) |
| Soil moisture | Grid (1 km) | https://csidotinfo.wordpress.com/data/global-high-resolution-soil-water-balance (accessed on 10 April 2022) |
| Lithology | Polygon | https://www.uni-hamburg.de (accessed on 6 April 2022) |
| Land use | Grid (1 km) | https://www.resdc.cn (accessed on 6 April 2022) |
| Geological environment division | Polygon | https://geocloud.cgs.gov.cn/#/home (accessed on 15 April 2022) |

(1)　Fixed-Effect Factors

Topography, road density, hydrology, and geological properties are the fixed-effect factors discussed here. The slope is the most widely adopted parameter for mass movement susceptibility mapping [7]. It is not only a prerequisite for the occurrence of a mass movement but also affects the infiltration process and the resulting field distribution [66]. Aspect represents the orientation of slope, which will affect radiation absorption, rainfall runoff and weathering conditions, thus indirectly influencing the occurrence of mass movement [67]. Profile curvature and plan curvature indicate the change rate of slope along and perpendicular to slope gradient, which primarily influences soil erosion and surface runoff [68]. These terrain factors were generated from Shuttle Radar Topography Mission (SRTM) DEM at 90m resolution [69].

Road density is important, as many mass movements tend to occur along the roads in mountainous areas. This is primarily due to the instability of the slope caused by the destruction of mountains for road construction. Thus, road density is a commonly utilized anthropogenic variable for assessing mass movement susceptibility [70].

Hydrological conditions, including river density and soil moisture, are important in mass movements. Banks of rivers may collapse due to infiltration of pore water and erosion of slopes [71]. The soil moisture plays a crucial role in soil cohesion and permeation, leading to changes in soil shear strength [72]. Data on roads and rivers are generated from the National Platform for Common Geospatial Information Services. Data on soil moisture is derived from the Global High-Resolution Soil-Water dataset at 1km resolution and calculated as an annual average value [73].

Concerning the geological properties, lithology was selected to represent the physical and chemical properties of rocks. There are 16 types of lithology in China, namely, Basic Volcanic Rocks (VB), Intermediate Volcanic Rocks (VI), Acid Volcanic Rocks (VA), Basic Plutonic Rocks (PB), Intermediate Plutonic Rocks (PI), Acid Plutonic Rocks (PA), Metamorphic Rocks (MT), Evaporites (EV), Pyroclastic (PY), Carbonate Sedimentary Rocks (SC), Siliciclastic Sedimentary Rocks (SS), Mixed Sedimentary Rocks (SM), Unconsolidated Sediments (SU), Ice and Glaciers (IG), Water Bodies (WB), and No Data (ND). The lithology data was derived from the Global lithological map (GLiM) developed by Hartmann and Moosdorf [74].

(2)　Random intercept factors

Next, we discuss random intercept factors associated with spatial heterogeneity of mass movement completeness, introduced into mixed-effects models as random intercept terms. Based on previous research, land use and division of the geological environment were chosen in China to account for mass movement incompleteness [57]. As for land use, mass movements in agricultural land or along transportation infrastructure are more likely to be blurred or removed by human activities. Those in forest areas are easily detected due to their distinct characteristics that differentiate them from the surrounding environment [54]. Thus, the mass movement inventory is expected to be underrepresented in arable land and overrepresented in forests. In this study, the land use data is derived from the Remote Sensing Monitoring Database of China's Land use/Cover in 2005 with 1km resolution, which was reclassified into five categories to better account for the bias of mass movement inventory. These are classified as Arable land (Ar), Forest land (Fo), Meadowland (Me), Settlements and Artificial land (SA), and Unutilized land (Un).

The China Geological Survey describes the geological environment as seven divisions based on geologic structure, geographical conditions, and geomorphology. These are the Northeastern plain and mountain region (I.NE), the Huang–Huai–Hai–Yangtze River Delta plain region (Yangtze), the South China low mountain and hill region (III.SC), the North China Loess Plateau region (IV.Loess), the Southwest karst mountain region (V.SW), the Northwest mountain and basin region (VI.NW) and the Qinhai–Tibet Plateau region (VII. Tibet) (Figure 1). Due to differences in topographical conditions and economic development, the completeness of mass movement inventory vary from region to region [57]. For example,

mass movement investigations are generally more detailed in the economically developed SC region. However, the Tibet region is too high to reach, resulting in poorer availability of mass movement [75].

There is no universal guideline on the choice of mapping unit for mass movement susceptibility assessment [76]. This paper selected the most popular grid with 1 km resolution as the primary mapping unit. All the thematic layers were resampled or converted to 1 km resolution for consistency. Since the mass movement inventory was stored as points, a grid was considered as one with mass movement if it contained at least one event of mass movement. If the grid unit did not include any mass movements, it was considered as one without any mass movements. Ultimately, a total of 90,558, 47,057 and 25,425 grid units were confirmed, which included landslide, rockfall and debris flow, respectively. The distribution maps of all influencing factors are shown in Figure 3. Further, it has been noted that the selection of ratio between absence data and presence of data and the sampling method will affect the accuracy of the mass movement susceptibility model [41]. Based on previous research, this study adopted a usual 1:1 ratio and performed random sampling on absence grid units [42,77].

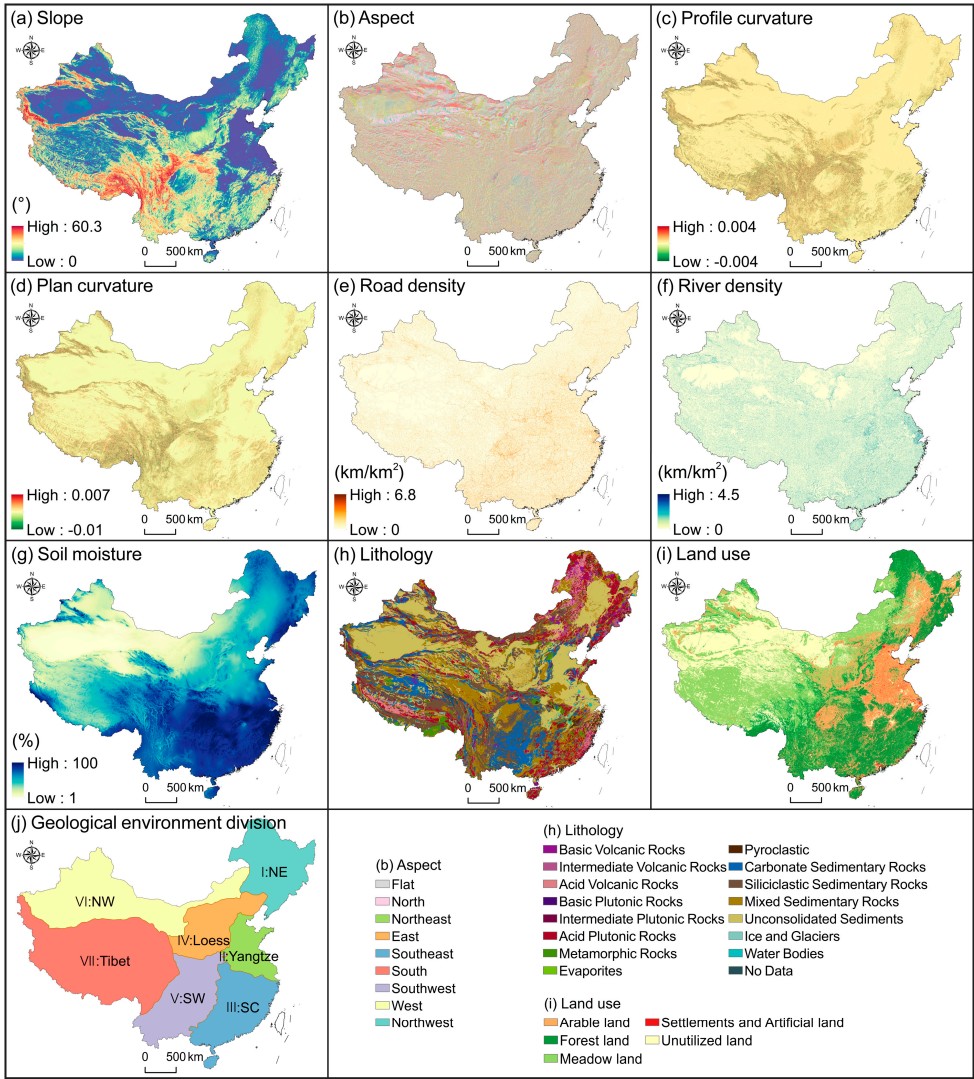

**Figure 3.** Spatial distribution map of influencing factors. (**a**) slope; (**b**) aspect (**c**) profile curvature; (**d**) plan curvature; (**e**) road density; (**f**) river density; (**g**) soil moisture; (**h**) lithology; (**i**) land use (**j**) geological environment division.

## 3. Methodology

The methodological flowchart of this study, including four main phases, is shown in Figure 4. In step 1, a spatial database was constructed with three mass movement inventories as response variables, eight basic predictors as fixed effects and two factors closely related to incompleteness of inventories as random intercepts. In step 2, preliminary data analysis was carried out to study the correlations between influencing factors and their relationships with mass movements, and to further confirm the factors that describe the incompleteness of the inventories. In step 3, three mixed-effects models were implemented for all the mass movements based on the established spatial database. It should be noted that the model fitting is based on fixed and random effects. In contrast, model prediction uses only fixed-effect factors, and the random effects that account for mass movement incompleteness are zeroed (i.e., averaged-out). In step 4, the prediction results of the three mixed models are compared and analyzed from both quantitative and qualitative perspectives.

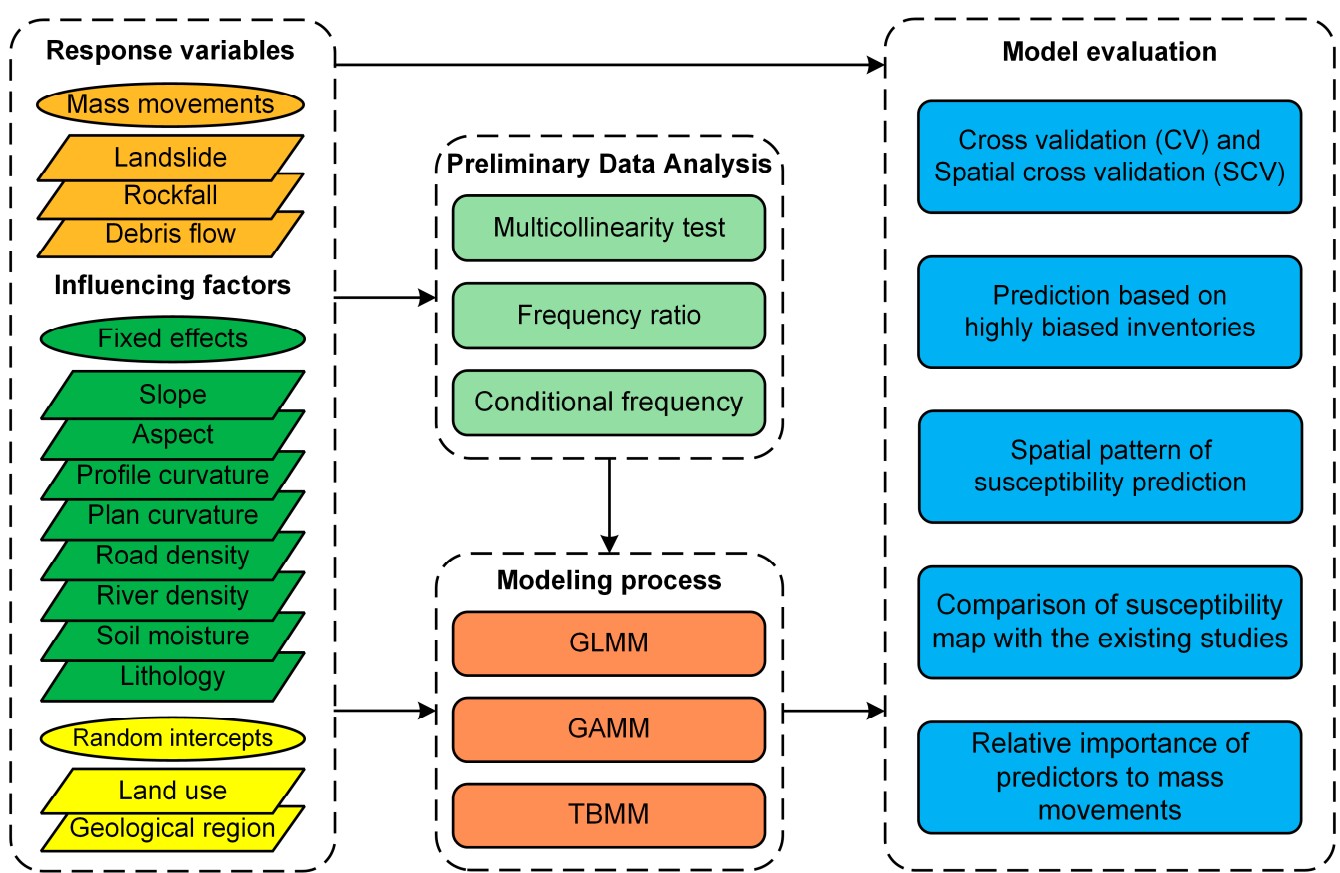

**Figure 4.** Workflow performed in this study.

### 3.1. Generalized Linear Mixed-Effects Model

The generalized linear mixed-effects model (GLMM) [78] combines and extends the characteristics of the linear mixed-effects model (LMM) and generalized linear model (GLM). Its dependent variable is no longer required to follow a Gaussian distribution (from GLM), and independent variables can contain both fixed and random effects (from LMM). GLMM has distinct advantages for analyzing the clustered, longitudinal and hierarchical data that are grouped at different levels [79]. In recent years, GLMM has been used to address data incompleteness in mass movement susceptibility mapping with considerable success [54]. Specifically, while modeling mass movement susceptibility, the Logit Link Function is adopted to represent the probability of mass movement occurrence P (Y=1) while specifying variables related to mass movement incompleteness as random intercept

terms and the other basic predictors as fixed-effect terms. The GLMM for mass movement susceptibility can be expressed as follows:

$$logit(P(Y = 1)) = \beta_0 + \beta_1 x_1 + \beta_2 x_2 + \cdots + \beta_m x_m + \gamma \tag{1}$$

where $\beta_0$ is the coefficient for the intercept, $\beta_1 \cdots \beta_m$ are the fixed-effect coefficients of associated influencing factors, $x_1 \cdots x_m$, and $\gamma$ is the random intercept component that is presumed to be normally distributed with mean zero and variance $\sigma^2$ [80]. For more description on the application of GLMM in mass movement susceptibility, see Steger et al. [54].

### 3.2. Generalized Additive Mixed-Effects Model

The generalized additive mixed-effects model (GAMM) [81,82] extends the properties of the GLMM approach by replacing linear functions with smoothing functions to allow nonlinear associations between the dependent and independent variables while maintaining additivity [83]. GAMM is highly flexible and can easily control overfitting because it employs non-parametric additive functions to model linear or nonlinear covariate effects [84]. To evaluate a binary outcome (i.e., mass movement presence/absence) in mass movement susceptibility modeling, the GAMM has the form:

$$logit(P(Y = 1)) = \beta_0 + f_1(x_1) + f_2(x_2) + \cdots + f_m(x_m) + \gamma \tag{2}$$

where the feature function, $f_1 \cdots f_m$, is constructed by a non-parametric smoothing spline, which can automatically model nonlinear associations without manually trying out many different transformations on each factor. Other parameters are similar to those in GLMM (Section 3.1). See Lin et al. [57] for more details on the methodology.

### 3.3. Tree-Boosted Mixed-Effects Model

Tree-boosted mixed-effects model (TBMM) [59] is a novel machine learning-based mixed-effects model that combines gradient-boosted trees with random effects. The model allows the relaxation of the linearity assumption of the response variable in a flexible nonparametric manner, and it can handle both continuous and discrete independent variables. The boosted trees have recently attracted significant attention in mass movement susceptibility mapping because of their state-of-the-art prediction performance and higher flexibility than other machine learning methods [8,85]. TBMM also has these advantages and can combine random effects to analyze grouped data; thus, it has great potential to solve the problem of incompleteness in mass movement susceptibility mapping. The equation of TBMM in this research is as follows:

$$logit(P(Y = 1)) = TB(X) + \gamma \tag{3}$$

where $X$ is the $m$-dimensional fixed-effects design matrix, e.g., there are $m$ predictor variables. $TB()$ is a boosted tree. More specifically, the GPBoost algorithm is adopted to train the model, which iteratively learns the (co)variance parameters of the random effects and uses a gradient boosting step to add a tree to the ensemble of trees. In particular, the LightGBM [86] library is used to learn tree-boosting. LightGBM is very suitable for handling large-scale data due to its higher efficiency than other gradient boosting trees. More detailed principles of TBMM can be found in Sigrist [59]. Regarding the hyperparameter settings of this study, we used the built-in grid search function ('gpb.grid_search_tune_parameters') of the "gpboost" package to select the optimal algorithm parameter. The final parameters settings for the three types of mass movements are shown in Table 2.

**Table 2.** Parameter settings of TBMM for three types of mass movement.

| Parameter | Debris Flow | Rockfall | Landslide |
|---|---|---|---|
| learning_rate | 0.8 | 0.5 | 0.8 |
| max_depth | 10 | 6 | 5 |
| min_data_in_leaf | 80 | 30 | 30 |
| num_boost_round | 200 | 200 | 300 |

*3.4. Model Evaluation*

To quantitatively assess the predictive ability of all the mixed-effects models, the Area Under the Receiver Operating Characteristic curve (AUROC) was adopted [87]. The value ranges from 0.5 to 1, where 0.5 is a random prediction and 1 is a perfect prediction. The AUROCs for all models were calculated by repeated Spatial Cross Validation (SCV) and non-spatial Cross Validation (CV) approaches [88]. In this context, 10-times-repeated 10-fold partitioning of training and testing sets were used for both SCV and CV. The final spatial mass movement susceptibility maps produced from each model have been checked for plausibility, considering the incompleteness of the data on spatial variation of the mass movement and the comparative analysis of the spatial pattern observed from different models.

All the data preprocessing was performed in ArcMap software. Before introducing the data into the model, a multicollinearity test and frequency ratio analysis was performed using python and ArcMap software. The conditional frequency plots were based on the cdplot function in the R. GLMM and GAMM were achieved with "lme4" [89] and "mgcv" [90] packages in software R, respectively. TBMM is built based on the package "gpboost" [59] in python. SCV and CV for GLMM and GAMM were estimated using R package "sperrorest" [88], and for TBMM, they were implemented with the "sklearn" [91] package in python.

## 4. Results

The results of this study mainly include analysis of influencing factors, quantitative performance comparison of different mixed models, spatial pattern comparison of susceptibility map, and evaluation of the relative importance of influencing factors to three types of movements.

*4.1. Preliminary Data Analysis*

4.1.1. Multicollinearity Test

Multicollinearity is a common issue in model evaluation. It occurs when there are high correlations among predictors, which can reduce the stability of the model or even cause the model to fail. Thus, it is essential to perform a multicollinearity calculation before the predictors are entered into the model. The tolerance ($TOL = 1 - R^2 J$) and variance inflation factor ($VIF = 1/TOL$) are typical indicators for testing multicollinearity. If the value of $TOL < 0.1$ or $VIF > 10$, it means serious multicollinearity [92]. The results demonstrate that the $VIF$ values of all predictors are $< 5$, and the $TOL$ values are $> 0.2$, among which the soil moisture achieved the highest $VIF$ of 4.3645 and the lowest $TOL$ of 0.2291 (Table 3). These results indicate that there is no multicollinearity problem for all selected predictors.

**Table 3.** Results of multicollinearity test for predictors.

| Influencing Factors | VIF | TOL |
|---|---|---|
| Slope | 2.3926 | 0.4179 |
| Aspect | 3.1363 | 0.3188 |
| Profile curvature | 1.1988 | 0.8342 |
| Plan curvature | 1.1532 | 0.8672 |
| Road density | 1.2351 | 0.8087 |
| River density | 1.3272 | 0.7535 |
| Soil moisture | 4.3645 | 0.2291 |
| Lithology | 2.3306 | 0.4291 |

### 4.1.2. Correlation Analysis between Mass Movements and Influencing Factors

The Frequency Ratio (FR) approach was adopted to analyze the association between inventoried mass movements and each conditional factor [93]. A FR value below 1 reveals a weak relationship, and a value above 1 indicates a high probability of mass movement occurrence [94]. Since FR can handle only categorical variables, the Jenks natural breaks method is adopted to classify continuous factors. Figure 5 depicts the relationship between the three types of movements and each influencing factor. It can be conclueded that the slope of the land illustrated a positive association with debris flows. However, for rockfalls and landslides, there was an initial increasing trend followed by a decreasing trend. Except for the first class of all mass movements and the last class of landslides, all other FR values exceed 1, indicating that the slope is crucial to the occurrence of mass movements. For aspect, the frequencies of three types of mass movements are slightly higher in east and southeast. For profile curvature, all movements have a high frequency of occurrence in the first three categories. Rockfalls also have a higher incidence in the class of 0.002–0.02, and landslides additionally have a higher incidence in the categories of 0.002–0.02 and 0.02–0.03, indicating that most mass movements are more abundant in concave and some convex areas. Regarding the plan curvature, mass movements are more frequently distributed within concave and convex classes. The results of FR values for road density and river density show that the occurrence frequency of mass movements is positively correlated with the density of roads and rivers, indicating their critical influence on mass movement occurrence. As for soil moisture, the FR values of debris flows initially increased and then decreased, while the frequency of rockfall and landslide occurrence is proportional to soil moisture. At the same time, debris flows have a higher frequency in areas with moderate humidity (categories 3 to 6). In comparison, rockfalls and landslides have a higher frequency in areas with high humidity (the last two categories). The lithology results show that Intermediate Volcanic Rocks, Basic Plutonic Rocks, Acid Plutonic Rocks, Metamorphic Rocks, Pyroclastic, Carbonate Sedimentary Rocks, Siliciclastic Sedimentary Rocks and Mixed Sedimentary Rocks have higher FR values, demonstrating a high likelihood of mass movement in these lithological units.

The three types of movements are more concentrated in arable and forest land for the land use factor. In addition, debris flows are abundant on meadowland. Landslides are also more distributed in settlements and artificial land. This phenomenon may be related to the discrepancies in the completeness of the inventory of mass movements for different land use types reported by Petschko et al. [95]. In addition, as can be found in Figure 6a, the distribution of land use varies between different slopes. Both arable land and settlements and artificial land are more frequently spread over gentler areas, while forest land and meadowland tend to be concentrated in steeper terrains. Therefore, this confounding relationship needs to be avoided when modeling. In the case of geological environment division, the distribution of mass movements in different regions varies greatly. Debris flows are reported in more detail in SW, Loess and Tibet regions, rockfalls are predominantly reported in SC and Loess regions, and landslides are more concentrated in SW and SC regions. The distribution difference in mass movements across the geological environment division is believed to be tightly associated with the difference in resource investment for mass movement investigations and the availability of mass movement information [57]. In addition, geological environment division and soil moisture are also spatially correlated (Figure 6b). For example, SC and SW are mainly distributed in humid areas, while NW is primarily located in arid areas. Therefore, this discrepancy in the representation of movements data within different geological environment subregions could have biased effects on soil moisture.

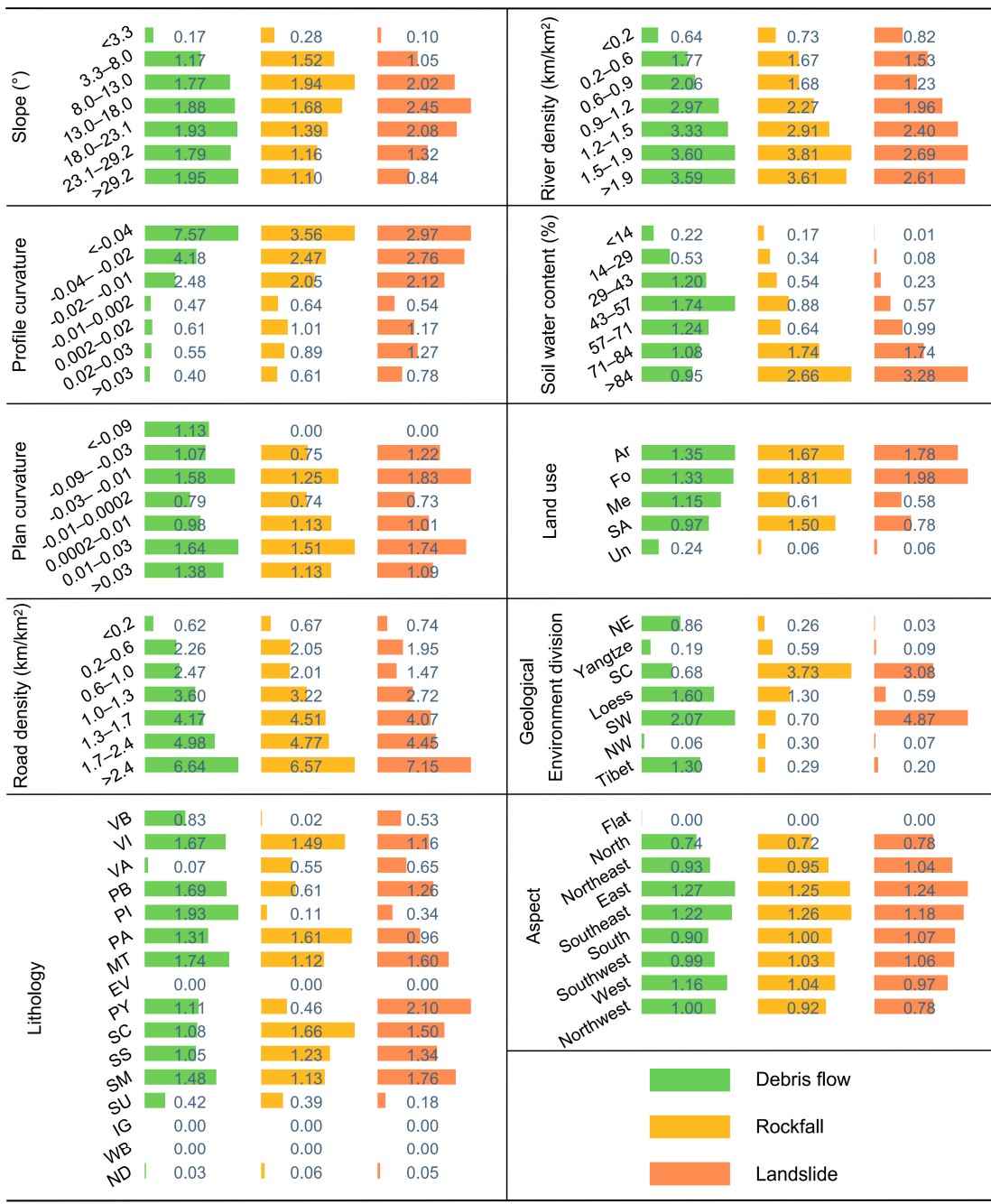

**Figure 5.** Correlations between three types of mass movements and influencing factors.

In summary, we find that land use and geological environment division are directly related to the incompleteness of data on mass movements. They are also associated with other predictors (e.g., slope and soil moisture). Therefore, mixed-effects models are needed to account for these biases [54,57].

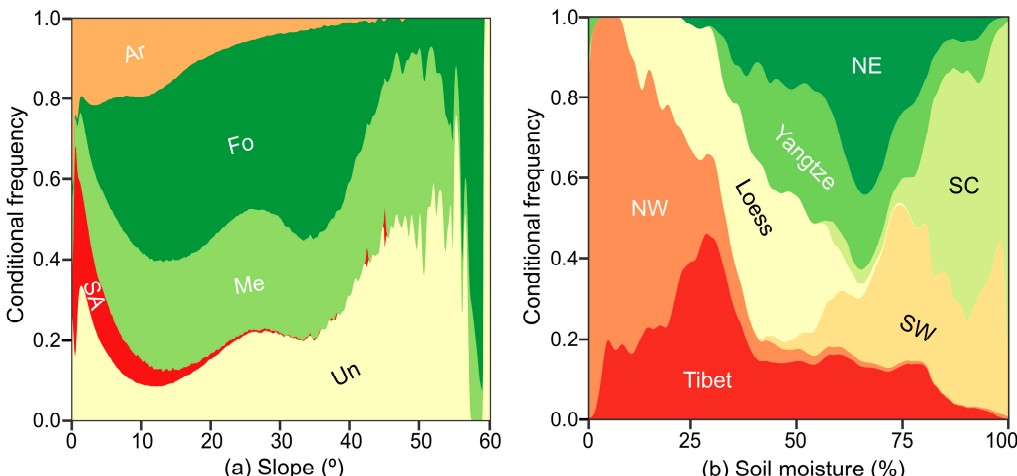

**Figure 6.** Conditional frequency plots between (**a**) land use and slope; (**b**) geological environment division and soil moisture.

### 4.2. Quantitative Performance Comparison

#### 4.2.1. Cross-Validation Results

Table 4 presents the comparison results of three mixed-effects models for different types of movements. For debris flow, the median AUROCs of non-spatial cross-validation (CV) for all models are higher than 0.8. For GAMM and TBMM, the median AUROCs of spatial cross-validation (SCV) are above 0.8. This value is lower than 0.8 for GLMM, indicating the better spatial and non-spatial performance of GAMM and TBMM. In addition, the median AUROCs of all models decrease (SCV vs. CV) are below 0.02, with all interquartile ranges below 0.1, demonstrating the robustness of spatial predictions. Regarding the rockfall, the median AUROCs of SCV results for all models are above 0.8, and the median AUROCs of CV results for GAMM and TBMM are higher than 0.8 but only 0.773 for GLMM, suggesting that GAMM and TBMM perform better. In addition, the median AUROCs of the SCV for GLMM and GAMM are abnormally greater than that of CV, and the SCV of TBMM is only 0.013 lower than CV with the interquartile ranges less than 0.15, indicating that the prediction performance of TBMM is more stable. In the case of landslide, the median AUROCs of CV results for all models are above 0.8, while the median AUROCs of SCV results for GLMM and GAMM are below 0.8, indicating their unstable spatial predictive performance. For TBMM, the median AUROC value dropped by (SCV vs. CV) only 0.018, indicating that its predictive power is more spatially robust compared to GLMM (0.039) and GAMM (0.056). Regarding the differences in interquartile, GAMM has the best performance with an interquartile of 0.055, and both GLMM and TBMM performed somewhat poorly, with interquartile ranges of about 0.19.

Overall, TBMM consistently produced SCV and CV results above 0.8 and higher than GLMM and GAMM for all types of movements, and its reduced values (SCV vs. CV) and interquartile ranges were acceptable, which are generally considered to reflect superior validation performance [54].

**Table 4.** Cross-validation results of three mixed models for mass movements.

| Model | AUROC<br>Median (1st–3rd Quantile) | Debris Flow | Rockfall | Landslide |
|---|---|---|---|---|
| GLMM | Non-spatial Cross Validation | 0.816 (0.813–0.819) | 0.773 (0.769–0.775) | 0.827 (0.825–0.830) |
| | Spatial Cross Validation | 0.799 (0.760–0.832) | 0.805 (0.679–0.817) | 0.788 (0.654–0.845) |
| GAMM | Non-spatial Cross Validation | 0.848 (0.844–0.852) | 0.801 (0.799–0.805) | 0.839 (0.836–0.842) |
| | Spatial Cross Validation | 0.844 (0.781–0.855) | 0.805 (0.734–0.846) | 0.783 (0.751–0.806) |
| TBMM | Non-spatial Cross Validation | 0.866 (0.863–0.868) | 0.830 (0.826–0.833) | 0.841 (0.837–0.844) |
| | Spatial Cross Validation | 0.848 (0.800–0.858) | 0.817 (0.733–0.865) | 0.823 (0.678–0.867) |

#### 4.2.2. Predictions Based on Highly Biased Inventories

According to Steger et al. [54], an excellent model could maintain higher predictive performance even when the inventory is severely incomplete. This study simulated several inventory data that were highly biased by randomly deleting 80% of debris flows, rockfalls and landslides in the SC and SW regions, NW and Tibet regions, forest land, and arable land, respectively. Finally, SC-and SW-related, NW- and Tibet-related, forest-related, and arable-related biased inventories were obtained (Table 5).

**Table 5.** The number of mass movements in different highly biased inventories.

| Movements | Original Data | SC and SW Regions | NW and Tibet Regions | Forest Land | Arable Land |
|---|---|---|---|---|---|
| Debris flow | 25,425 | 19,307 | 17,252 | 19,393 | 20,218 |
| Rockfall | 47,057 | 22,479 | 41,697 | 31,632 | 35,192 |
| Landslide | 90,558 | 28,187 | 85,622 | 58,112 | 66,092 |

Then, all mixed models were trained based on these highly biased data and were used to predict the unmodified mass movements; the results are shown in Figure 7. A conclusion can be drawn that TBMM exhibits the best predictive performance for all the cases. At the same time, the range of variation of the prediction results based on different biased data was compared. For debris flow, all models showed a small range of variations from 0.002 for GAMM, 0.003 TBMM and 0.006 for GLMM. In the case of rockfall, the value for TBMM remained stable with a range of 0.007, while the GAMM and GLMM performed poorly with values, respectively, in the ranges of 0.020 and 0.023. Regarding landslide, GLMM and TBMM performed best with a range of 0.006, while GAMM was the worse, with a range of 0.025. Overall, TBMM showed the best score with a stable variation range (less than 0.01) across all types of land movements, indicating its advanced and robust prediction performance based on highly biased inventories.

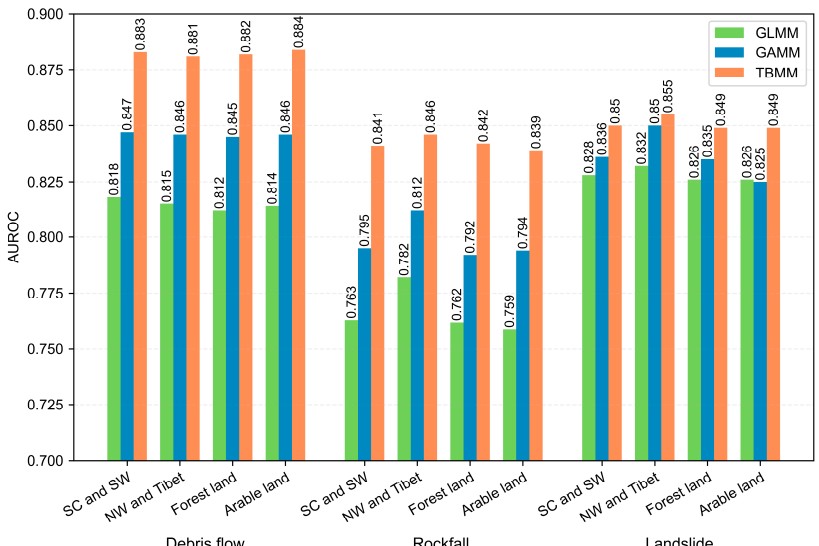

**Figure 7.** Comparison of model predictions (AUROC) for different types of mass movements based on highly incomplete data.

#### 4.3. Spatial Pattern Comparison of the Susceptibility Map

Susceptibility maps for three movements were generated by predicting the probability of occurrence for each grid using three mixed-effects models separately. Then, the susceptibility index was classified into five categories using the Jenks natural break strategy in ArcMap 10.8 [96]. Figures 8–10 demonstrate the spatial patterns of debris flow, rockfall and landslide susceptibility based on the three mixed models and detailed local area

comparisons. Regarding the differences between susceptibility maps, we first quantified susceptibility levels from very low to very high as one to five, and then the differences are obtained by subtracting one susceptibility map from the other through Raster Calculator tool in ArcMap. The results of debris flow show that three susceptibility maps (Figure 8a–c) are generally consistent. Areas that are highly prone to mass movements are abundant in the hilly areas of the Changbai Mountains (NE region), the southeast hills (SC region), and the Taihang Mountains (Loess region). The mountains around the Sichuan Basin (SW region), the Altai Mountains and the Tian Shan Mountains (NW region) and the mountains of the Qinghai–Tibetan Plateau (Tibet region) are also highly prone to mass movements. These results show that the three mixed-effects models can effectively reduce the impact of inventory bias, so they also have better predictive capability in the NW and Tibet regions, where there is a significant lack of data.

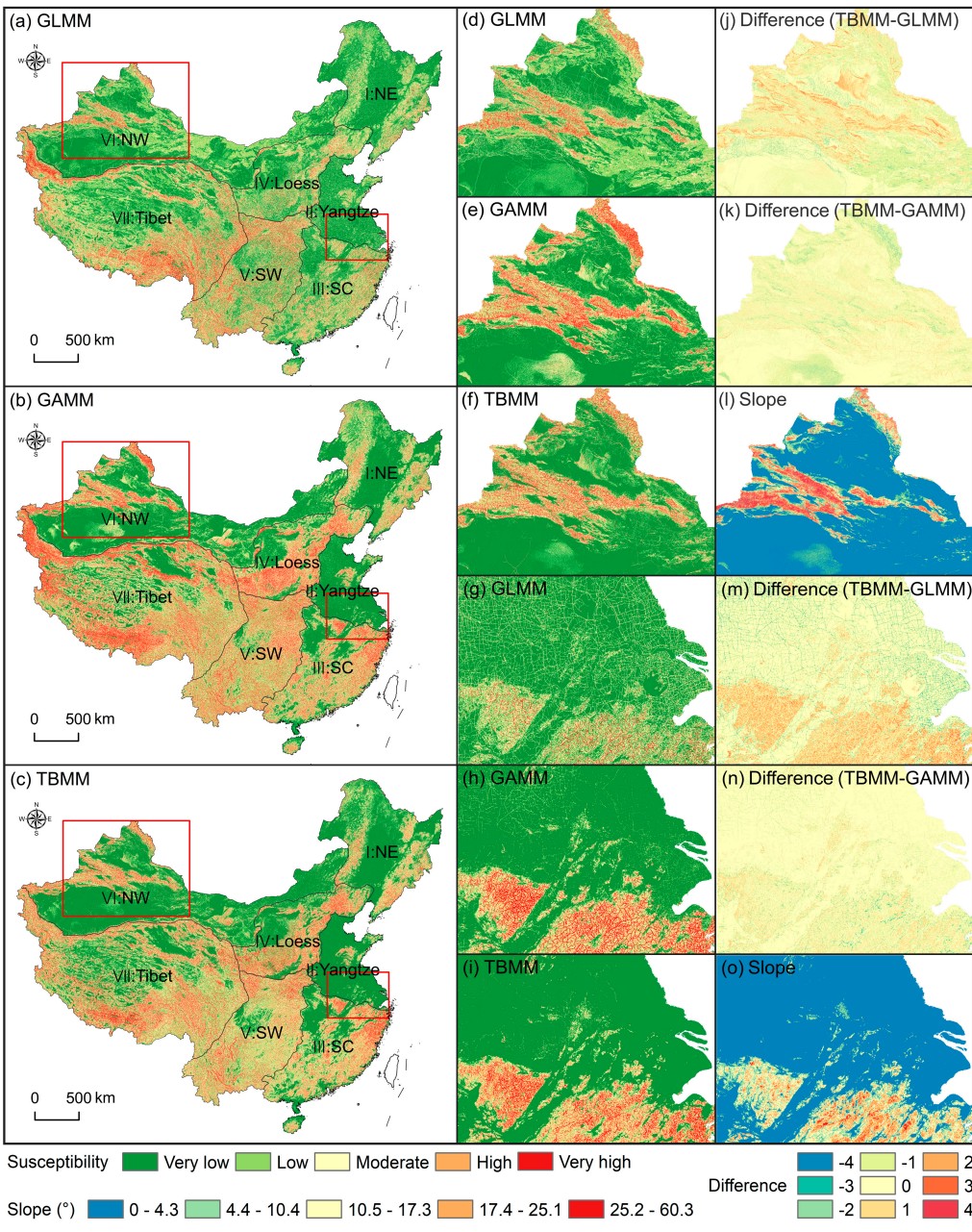

**Figure 8.** Debris flow susceptibility maps. (**a–i**) Susceptibility maps for China and local areas yield by three mixed models; (**j,k,m,n**) the difference between TBMM and other models in local areas; (**l,o**) the topography slope in local areas.

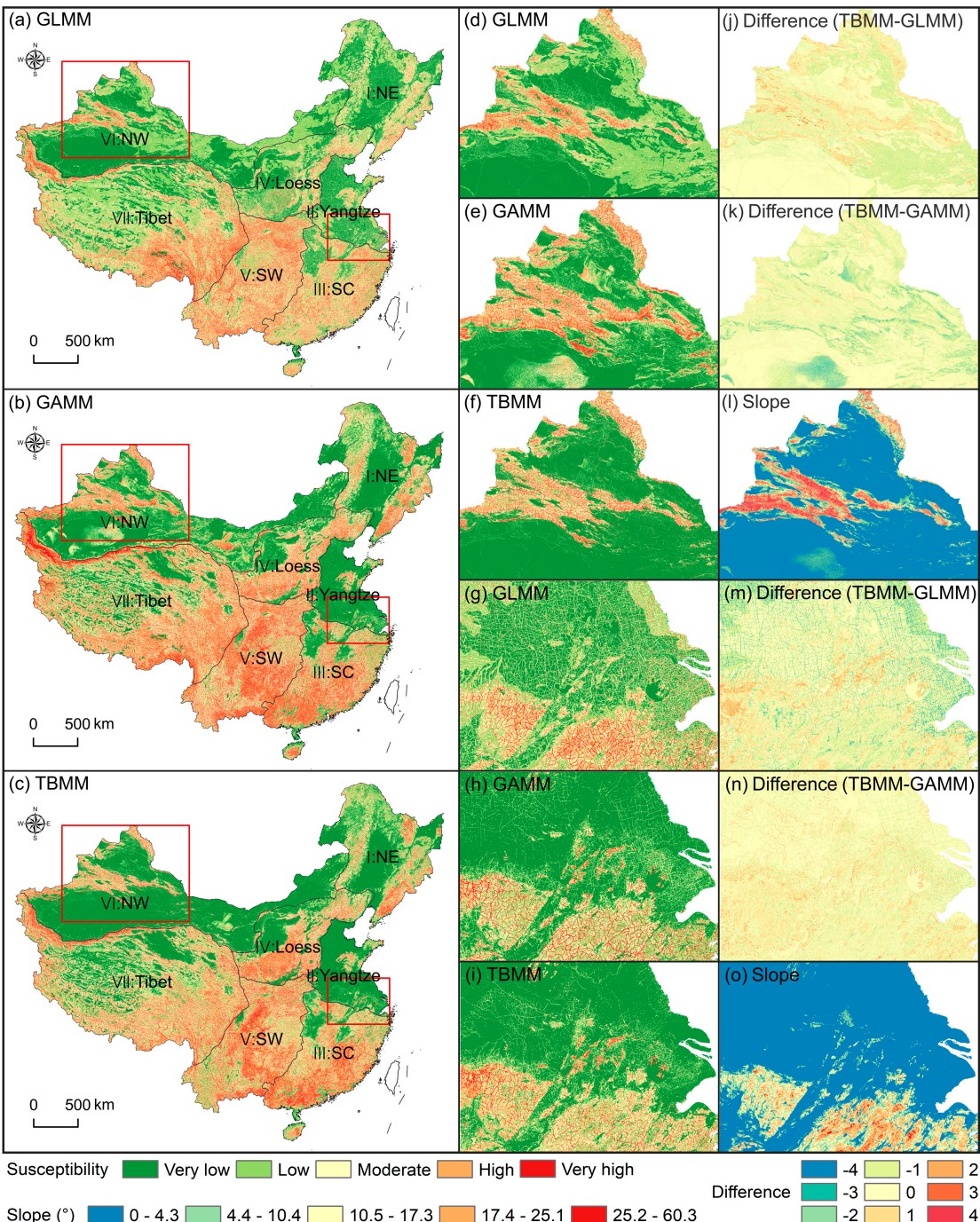

**Figure 9.** Rockfall susceptibility maps. (**a–i**) Susceptibility maps for China and local areas yielded by three mixed models; (**j,k,m,n**) the difference between TBMM and other models in local areas; (**l,o**) the topography slope in local areas.

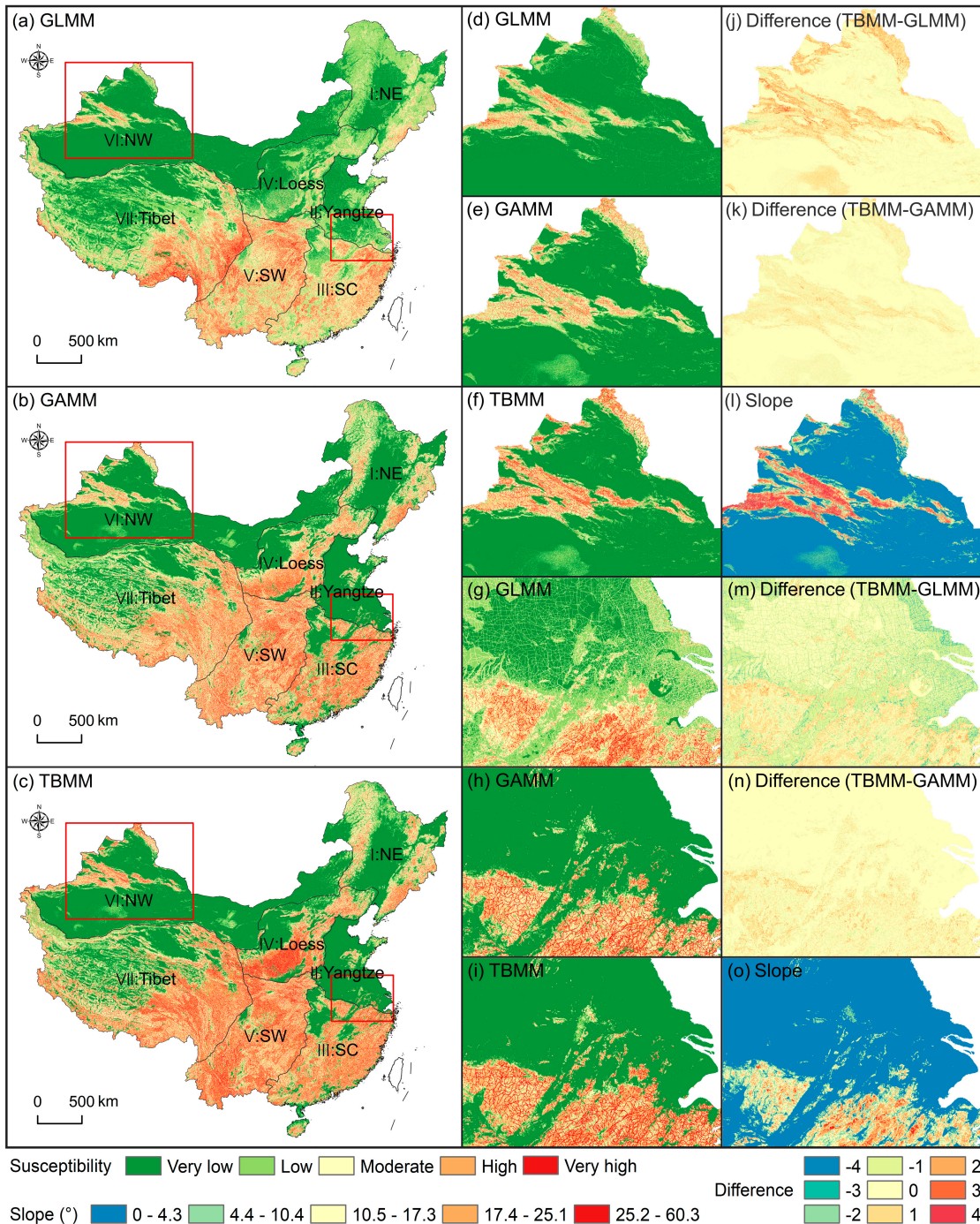

**Figure 10.** Landslide susceptibility maps. (**a–i**) Susceptibility maps for China and local areas generated by three mixed models; (**j,k,m,n**) the difference between TBMM and other models in local areas; (**l,o**) the topography slope in local areas.

Further, it has been observed that the susceptibility maps produced by GLMM are poor representations of the ground truth. For example, the slope angles in the Yangtze River Delta Plain are almost below 5 degrees. Still, the susceptibility map obtained by GLMM is greatly affected by roads in the plain. Many grids near roads are classified as medium or high susceptibility (see Figure 8o,g). The difference between TBMM and GAMM is lower than that of GLMM, with most values ranging between −1 and 1 (Figure 8j,k,m,n). Their predicted susceptibility in the plains was generally very low and significantly lower than that of GLMM (Figure 8h,i,m). The susceptibility was higher in mountainous areas than in

GLMM (Figure 8j,m), suggesting that the prediction results of TBMM and GAMM are more reasonable. Besides, some areas in the middle of the Taklimakan desert region (southern NW region) are predicted by GAMM to be moderate to high susceptibility (Figure 8e). In fact, the incidence of debris flow in arid regions is very low, and there are no debris flow points in the region (Figure 2a). In contrast, the susceptibility based on TBMM is lower than that of GAMM in this region (Figure 8k), indicating that the prediction of TBMM is better.

For rockfall, the susceptibility results of the GLMM (Figure 9a) performed relatively poorly. For example, in the Yangtze River Delta Plain, GLMM is severely affected by roads, and many areas are classified into medium to very high levels (Figure 9g). Overall, TBMM (Figure 9c) and GAMM (Figure 9b) are similar and perform better than GLMM in both mountains (Figure 9j,m) and plains (Figure 9m). TBMM also performed better than GAMM in desert areas (Figure 9k). The results indicate that GAMM may have overestimated the susceptibility in many areas, while GLMM underestimated the results.

Regarding the landslide, both TBMM (Figure 10c) and GAMM (Figure 10b) showed better performance. GLMM classified many areas in the Yangtze delta plain as moderate-to-high landslide susceptibility types (Figure 10g), which obviously do not match the actual topographic features. In addition, it can be observed that TBMM generates higher susceptibility indices than GAMM in the Tianshan and Altai mountains and the southeastern hills (Figure 10k,n). These results are similar to a previous study that used a representative landslide inventory from the local area [97]. The comparability of results suggests that TBMM has a more robust predictive capability.

To understand the overall pattern of mass movement distribution and mass movement susceptible areas, we performed summary statistics on the distribution of mass movement in each susceptibility class, distribution of susceptibility classes and the frequency ratio of mass movements in each susceptibility class (Figure 11). In general, models with more historical mass movement points concentrated in predicted high-prone areas are considered to have better performance [98]. From Figure 11, it can be found that for the three types of mass movements, only TBMM's results have more than 80% of the mass movement points concentrated in the areas with high and very high susceptibility (81.6%, 80.4%, and 85.1% for debris flow, rockfall and landslide, respectively), and the least mass movements were classified to low and very low-prone areas (6.9%, 5.8%, and 2.5%, respectively). Additionally, according to the results of Figure 11c, with the improvement of the susceptibility grade, the frequency ratios of debris flow, rockfall and landslide points increased by: 199 (0.033–6.557), 123 (0.039–4.813) and 247 (0.017–4.192) times, respectively, in TBMM; 104 (0.048–4.993), 52 (0.077–4.015) and 190 (0.027–5.139) times, respectively, in GAMM; and 51 (0.135–6.934), 37 (0.123–4.493) and 134(0.037–4.962) times, respectively, in GLMM. Therefore, the susceptibility maps produced by TBMM showed the most reliable results. In order for readers to accurately determine the susceptibility of any location, we provide the original data of three susceptibility maps generated by TBMM in the Supplementary Materials.

Almost all previous literature has only evaluated landslide susceptibility in China. Therefore, we compare the TBMM-generated landslide susceptibility map from this paper with three previous studies carried out in recent years (Figure 12). In general, Figure 12a is very similar to Figure 12b, which also employed a mixed-effects model. The results in both these figures predicted higher susceptibility in the NW and Tibet region's mountainous areas compared to those shown in Figure 12c,d. Both of these are the results from the application of traditional non-mixed models. Due to the high altitude and sparse population of Tibet and NW regions, the survey of landslides is relatively rough and possibly not representative of the actual situation. This example demonstrates how the lack of data significantly influences the model's outcome and confirms the mixed-effects models' superiority.

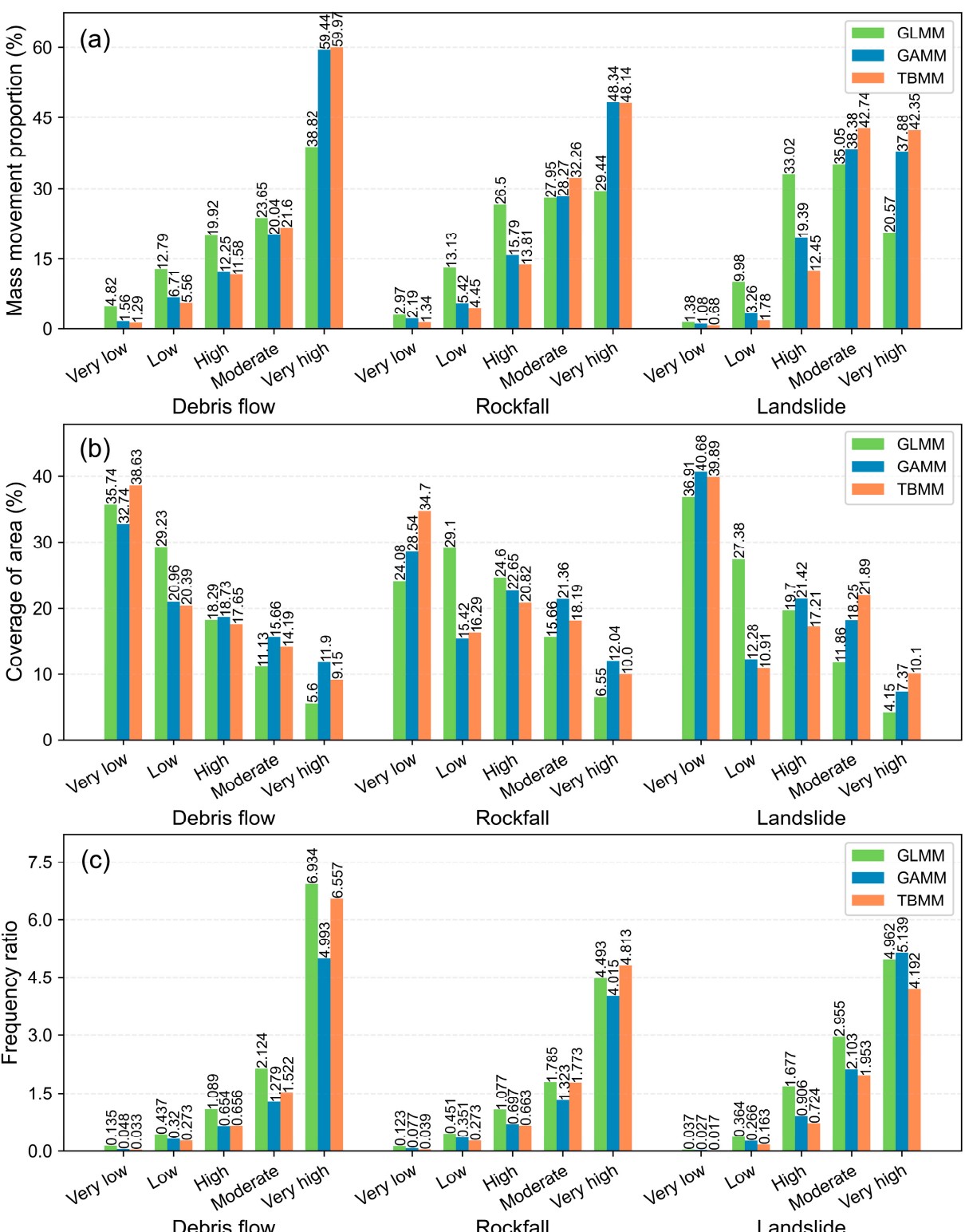

**Figure 11.** Statistics on the susceptibility maps of three type of mass movements. (**a**) distribution of mass movements in each susceptibility class; (**b**) distribution of susceptibility classes; (**c**) frequency ratio of mass movements in each susceptibility class.

Further, the results presented in Figure 12a show lower susceptibility in the Yangtze River Delta plain compared to the situation presented in Figure 12b. This result indicates that TBMM performs better in areas where landslides are unlikely to occur. It is to be noted

that between these two cases, there is a notable difference in the basic unit used for the study. While the spatial mapping unit used in Figure 12a is a grid with an area of 1 km², that for Figure 12b is a much larger sub-watershed, with an average area of 129.1 km². Thus, in the latter case, almost all the mountainous areas in the SW and SC regions are predicted to be more vulnerable, which can be misguiding and, therefore, not appropriate to be recommended as guidelines for the government for decision-making and evolving disaster prevention measures. On the contrary, the TBMM-based results that presented in Figure 12a are finer and more realistic, which allows for more targeted development of disaster prevention and resource investment strategies.

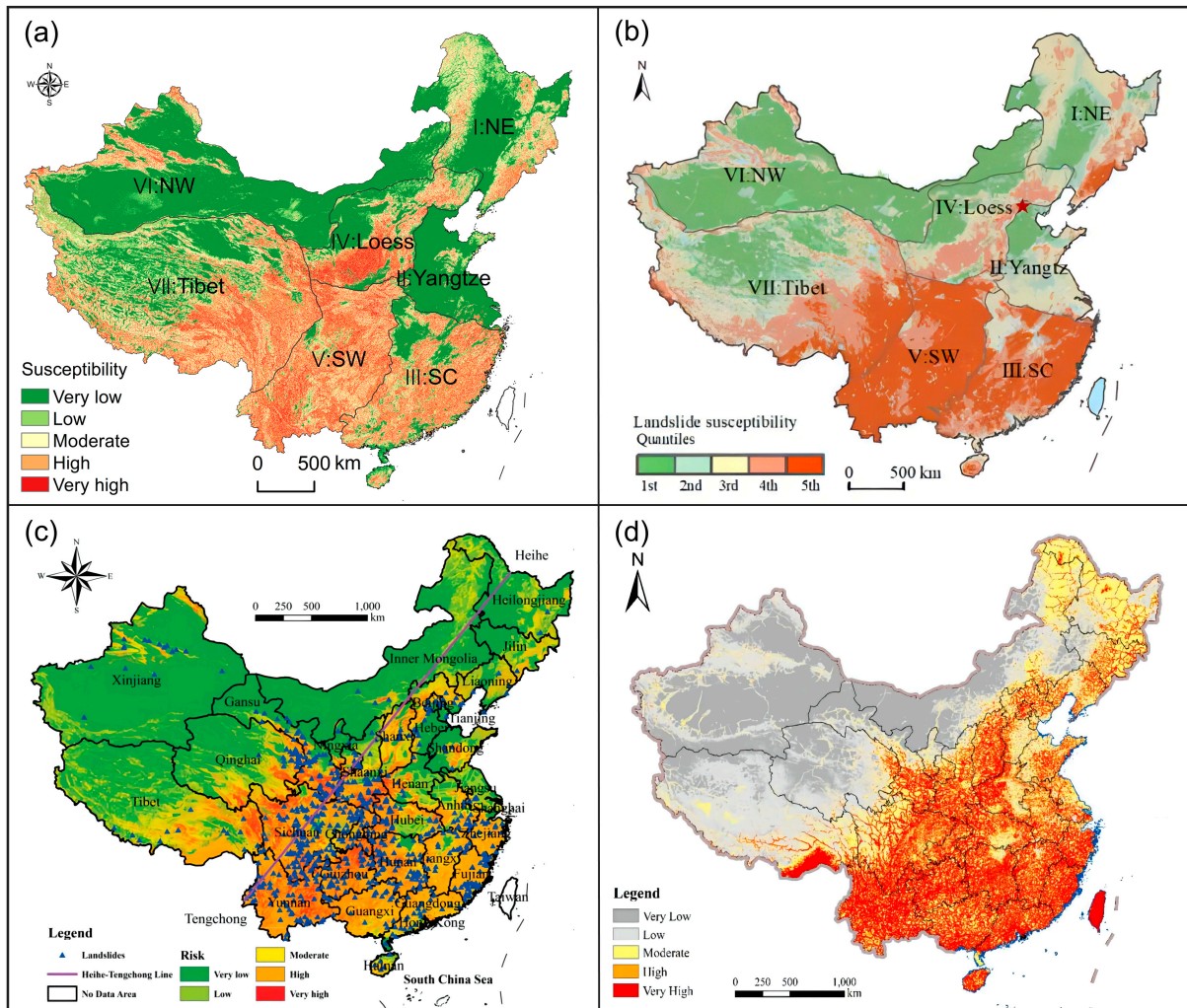

**Figure 12.** Comparison of results from this study with that of previous research. (**a**) Landslide susceptibility map generated by this paper based on TBMM; similar maps from (**b**) Lin et al. [57]; (**c**) Liu and Miao [99]; (**d**) Wang et al. [10].

### *4.4. Factor Contribution Analysis*

Since the fixed part of TBMM is fitted with LightGBM, the relative importance of each predictor for susceptibility modeling can be obtained. Results illustrating the mean and standard deviation (error bar) of the relative importance of each factor based on both spatial and non-spatial cross-validations are shown in Figure 13. It can be found that profile curvature, slope, road density and soil moisture are the four significant factors with relative importance greater than 0.1 for susceptibility modeling of debris flow. With relative importance of less than 0.05, aspect and plan curvature are factors that seem to have the least importance. In the case of rockfall, the slope, road density and soil moisture

contributed more to predicting rockfall risks with relative importance above 0.1. With a relative importance factor of less than 0.05, aspect, plan curvature and lithology have little influence on rock fall. In the case of landslide susceptibility modeling, the slope, road density and soil moisture have more influence, with their relative importance of more than 0.1. In contrast, the aspect and plan curvature seem relatively insignificant predictors.

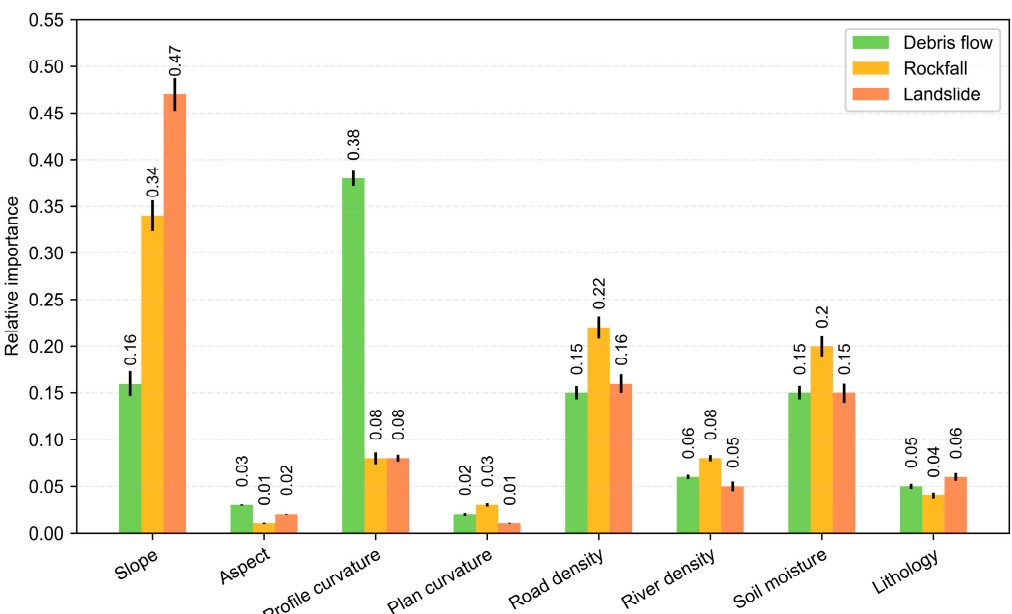

**Figure 13.** The relative importance of predictors in different mass movements.

The influencing parameters in the three types of movements have varying levels of importance. While slope, road density and soil moisture contribute significantly to the generation of debris flow, rockfall and landslide susceptibility, plan curvature and aspect appear to have the least importance in triggering all these movements. Additionally, it is noted that profile curvature has the highest importance in debris flow but is less critical for rockfall and landslide.

## 5. Discussion

With the development of artificial intelligence and the improvement in computing power, significant progress has been made in mass movement susceptibility modeling. An increasing number of susceptibility assessment models and optimization algorithms are being developed, and they are becoming increasingly complex [7–9]. However, reducing the bias propagation effect originating from the unavoidable incompleteness of mass movement inventory remains a fundamental and important challenge for susceptibility assessment, especially on a large scale such as that of China [57].

To address the issue of incomplete inventory in modeling mass movement susceptibility and to make realistic predictions, this research proposes three mixed-effects models, namely, GLMM, GAMM and TBMM. These models combine basic predictors (topography, road density, hydrology, geological properties, etc.) and random effect factors that account for biases in the mass movement inventory to improve the reliability of susceptibility evaluation in China. It is worth noting that all mixed models use both fixed and random effects to train the algorithm while only using fixed effects to predict the final susceptibility index. This step ensures that all models can counterbalance the adverse effects of biases in the inventory. However, there are some fundamental differences between the structure of the three models. The fixed part of GLMM is fitted by a parametric linear Generalized linear model (GLM). On the other hand, GAMM uses a semi-parametric nonlinear Generalized additive model (GAM) to fit the fixed effects term. Deviating from GLM and GAM, the fixed effects of TBMM are implemented by a non-parametric nonlinear LightGBM model.

Our study indicates that the forms of GAMM and TBMM are more flexible and may better fit the relationship between basic predictors and response variables. The empirical results also confirm this speculation. Based on the results presented in Table 4, GLMM had the worst performance in Cross-validation (CV) and Spatial cross-validation (SCV) for all types of mass movements. It is to be noted that with only half (3/6) of median AUROCs above 0.8, the level of performance of GLMM was low. While the median AUROCs produced by GAMM were mostly (5/6) above 0.8. TBMM performed best, with all values above 0.8. As for the difference between median AUROCs of SCV and CV, TBMM performed best, with the largest variation of 0.018, followed by GLMM with the largest difference of 0.039, and GAMM performed worst with the largest variation of 0.056. These differences are significantly more minor than those of the traditional non-mixed-effects model in the results of Lin et al. [57], and are close to the results of the mixed-effects model it used, which reflects the robustness of the mixed-effects model. Furthermore, according to Steger et al. [54], a good model can still have a high prediction score for the original mass movement data when the training data is highly biased. This paper simulated several types of highly biased inventory data to train all models and then predict the position of the original mass movements (Figure 7). It was found that TBMM still outperformed the other mixed models with all AUROCs above 0.8 and with the smallest fluctuations, further demonstrating the excellent performance of TBMM.

For the final mass movement susceptibility map generated for China, it can be found that both the mixed-effects models (Figures 8–10 and 12a,b) and the traditional non-mixed models (Figure 12c,d) yield similar high to very high susceptibilities in the SC, SW and Loess regions of China. It shows that the mass movement inventory in these areas is well represented; thus, all the models can yield good results. However, too low susceptibility levels for the northwestern mountainous areas (Tian Shan Mountainous and Altai Mountainous) obtained from traditional models are not credible, as they propagate the biases of mass movement inventory directly into the final results. On the other hand, results of all mixed-effects models classified the susceptibility of mountainous areas in the NW region as moderate to very high, which are also consistent with the results based on a representative inventory [97]. In addition, from the comparison of the susceptibility maps from different mixed effects models, it is found that TBMM not only maintained its excellent performance in mountainous areas but also did well in plain and desert areas, where mass movements are unlikely to occur. All these results indicate the superior quality of TBMM. Finally, the factor contribution analysis shows that the specific dominant factors of three type of mass movements are different (Figure 13). However, in general, topographic factors are the most important, followed by human activities and hydrology, indicating that topographical conditions are very crucial to the occurrence of mass movements. This observation is consistent with the views of most published results [42,64,65,100].

The results of this research emphasize the necessity to account for the effects of inherent inventory bias for susceptibility mapping, especially where the study area is as large as China. Considering this aspect of modelling, we implemented three mixed-effects models to account for the incompleteness of mass movement data, and our studies have suggested the superiority of TBMM. We find that this model can fit the relationship between the basic predictor and the response variables quite well while reducing the effect of the inventory bias. Therefore, this superior mass movement susceptibility evaluation model will provide a solid foundation for subsequent risk analysis and disaster prevention. In recent years, several other novel mixed-effects models have emerged. Among them are the MERF and BiMM combined with random forests [101,102]; the GMET and GMERT that combined decision trees [103,104]; and the MeNets and LMMNN combined with neural networks [105,106]. These novel mixed models have great potential to be explored in the field of mass movement susceptibility mapping. Moreover, there are some other strategies to deal with the incompleteness of the inventory data, such as using a fuzzy logic model to simulate the distribution characteristics of the entire mass movement based on a small portion of the mass movement inventory [107]. Training the susceptibility model for a

small area, where the inventory was considered relatively complete, and then making predictions for the whole research region based on the calibrated model [18]. Combining the results of remote sensing interpretation, heuristic and multinomial statistical models to compensate for the uncertainty caused by limited inventory data [75]. Applying the maximum entropy model to deal with limited data due to its advantage of not requiring negative samples [108,109]. A comparison of mixed-effects models with these methods should be considered in future studies.

Although this study found a better mixed-effects model to be more efficient in dealing with biased inventory, limitations still exist, starting with the size of the mapping unit. Due to the computational inefficiency of mixed-effects models, we need to find a compromise between resolution and computational efficiency in such a large study area. The 1 km × 1 km grid used in this study is relatively coarse, which may result in some grids containing more than one mass movement. More reasonably sized mapping units or intensity mapping instead of susceptibility mapping can be explored in future studies, as has been suggested by some previous researchers [40,110]. Secondly, implementing mass movement susceptibility mapping over such a large area makes it impossible to obtain an unbiased input data. Although the mixed-effects models we used can minimize this biasing effect, the results remained difficult to interpret and validate. Thirdly, this paper employed grid search to optimize the parameters of TBMM, but this method is still rough and extremely time-consuming [111]. Exploring better parameter optimization strategies is a focus of future research. Fourthly, Some studies have found that recent mass movements are more likely to affect future mass movements than earlier ones [112]. Since the mass movement inventory used in this paper have been collected since 2005, there may be some older mass movements that have little impact on the future and therefore affect the predictive performance of the model. Finally, we used road and river data in 2017 and land use data in 2005 due to limitations of data acquisition, which may influence the plausibility of the model predictions and need to be improved in future work. Given these limitations, we emphasize that the three types of mass movement susceptibility maps in this paper provide a general situational awareness of mass movement-prone areas in China, but they are not recommended for local decision making.

## 6. Conclusions

The incompleteness of inventory data is inevitable while performing the susceptibility mapping for mass movements in large areas such as China. The mixed-effects model proposed in recent years can solve this problem well, but there are also many ways to achieve it. In this paper, three mixed-effects models are implemented to evaluate the susceptibility of mass movements in China, and several important conclusions can be drawn.

(i)     From a quantitative point of view, the tree-boosted mixed-effects model (TBMM) performs best in both spatial and non-spatial cross-validation for all mass movements. In addition, when further reducing the completeness of inventory data in different categories of land use or geological environment division, TBMM maintained the best AUROC scores with little variation among the different highly biased types.

(ii)    From a qualitative point of view, the derived TBMM yielded more plausible spatial susceptibility patterns than the other two mixed models and conventional methods discussed in the existing literature.

(iii)   Through the factor contribution analysis, it was found that the profile curvature and slope contribute significantly to the evaluation of debris flow. For rockfall, slope, soil moisture and road density had more significant contribution. Regarding the landslide, slope and road density were the most critical factors.

In general, this paper aims to explore the best mixed-effects model to counterbalance the undesired effects of incomplete inventory data in susceptibility mapping and finally confirm the superiority of TBMM from both quantitative and qualitative perspectives. The susceptibility maps obtained in this study serve as a foundation for disaster prevention and

spatial planning. Further, this model provides a reference for future susceptibility mapping of other regions.

**Supplementary Materials:** The following supporting information can be downloaded at: https://www.mdpi.com/article/10.3390/rs14236068/s1, The original format of the three types of mass movements susceptibility maps generated by TBMM.

**Author Contributions:** Conceptualization, Y.Z.; methodology, Y.H.; validation, Y.Z. and Y.H.; formal analysis, Y.H.; resources, Y.Z.; data curation, Y.Z.; writing—original draft preparation, Y.H.; writing—review and editing, Y.H. and Y.Z.; visualization, Y.H.; supervision, Y.Z.; funding acquisition, Y.Z. All authors have read and agreed to the published version of the manuscript.

**Funding:** This research was funded by the National key research and development program of Ministry of Science and Technology (No. 2022YFF0711704), the National Cryosphere Desert Data Center (No. E01Z790201) and the Capacity Building for Cryosphere Desert Data Center, Chinese Academy of Sciences (No. Y929830201).

**Data Availability Statement:** All the data are available in the public domain at the links provided in the texts.

**Acknowledgments:** Thanks to the scholars who provided the data used in this work. Thanks to the editors and reviewers for their comments.

**Conflicts of Interest:** The authors declare no conflict of interest.

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
