# Peer review of "Comparison of Three Mixed-Effects Models for Mass Movement Susceptibility Mapping Based on Incomplete Inventory in China"

_remotesensing, doi:10.3390/rs14236068_

Round 1

Reviewer 1 Report

The issue the paper deals with is interesting but there are some moments which, in my opinion,should be clarified, Mathematical modeling of a physical phenomenon should be based on some crucial concepts. I think that geophysical data can be interpreted with the help of so called  joint inversion. Each geological or geophysical factor is to be considered separately and only after performing a comprehensive analysis a mathematical model of a phenomenon can be build.

I think that the spatial heterogeneity can not be regarded as a random factor. Coversely, this factor is very important when reconstructing the sources of physical fields. The main features of matrix (3) are to be given.  If there is a bad-conditioned linear algebraic equation system so a regularization algorithm is needed. The correlation analysis on Fig. 4 is very hard to perceive.  Besides, I can not infer an apparent correlation between the geological division and soil moisture from Fig.5a and 5b.

In addition, I can recommend the authors give more details on mathematical models used in this paper. Why do You prefer these 3 models? The nonlinear dependence of the most important factors should be described more accurately. 

Reviewer 2 Report

This manuscript (remotesensing-2026964) aims to compare three mixed-effect methods (namely generalized linear mixed-effects model (GLMM), generalized additive mixed-effects model (GAMM), and tree-boosted mixed-effects model (TBMM)) for geohazard (namely debris flows, rockfalls and landslides) susceptibility mapping based on incomplete inventory in China from 2005. Although it is an easy-to-follow manuscript, it is not entirely new to use these common mathematical methods in geohazard susceptibility mapping. Another very serious concern is that some related studies have been neglected. A further and detailed literature review must be conducted. Also, the current results of this study can hardly be reviewed because of those problems about data and methodology. Therefore, at least a “Major Revision” is required. My suggestions and comments are presented as follows:

- 1. Both the Abstract and the Introduction Section are weak because the authors did not clearly raise an important scientific question or gap related to geohazard susceptibility mapping. Therefore, potential readers can hardly identify the need that the authors should have to provide a new solution from an international perspective. What I have learned from the introduction is that the authors applied some previous established models at a national scale. Note that those mathematical techniques (namely the mixed-effect methods) are not new methods in geohazard susceptibility mapping.

- 2. In Line 49~53: the authors mentioned that: "The quality of the mass movement inventory; the selection of spatial mapping units and their resolutions [9] and the sampling strategy for mass movement-free units [10] are criteria that would affect the data quality". Yes, it is true. However, this manuscript did not deal with many of these problems.

- 3. The authors need to answer clearly why we must build the geohazard susceptibility models at the national scale? Actually, the influencing factors and spatial patterns will be very different across different provinces and regions because of the existence of significant spatial heterogeneity.

- 4. Section 2.2.2 Mass movement influencing factors: the selection of the influencing factors of mass movement is still not enough because many essential factors have not been considered. For example, it is unreasonable that "Precipitation and earthquakes were regarded as triggering factors and were therefore excluded". Actually, mass movement will be caused not only by local precipitation (at/over each grid cell), but also by flood water from high precipitation area in the upstream regions. In addition, aspect, soil type, soil depth, distances from human settlement, normalized difference vegetation index, etc. are also very important.

- 5. The authors need to answer clearly why the density factors are used, rather than the distance factors. For example, why the density of rivers and roads was used rather than the distance from rivers and roads.

- 6. Another serious concern is that the authors must look further into the latest research in this field. In fact, the literature review is far from enough. In particular, the maximum entropy (MAXENT) algorithm has been successfully used in geohazard susceptibility mapping. However, this well-accepted technique is totally ignored in the manuscript, and the following articles should be cited. The Introduction section is meant to set the context for your research work and highlight how it contributes to the knowledge in this field and builds on previous similar studies.

Predicting future urban waterlogging-prone areas by coupling the maximum entropy and FLUS model. Sustainable Cities and Society, 2022, 80: 103812

Land subsidence hazard modeling: Machine learning to identify predictors and the role of human activities. Journal of Environmental Management, 2019, 236.

Evaluation of multi-hazard map produced using MaxEnt machine learning technique. Sci Rep, 2021, 11

- 7. The authors need to answer clearly why those three mixed-effects methods were selected. For example, why not use the other more common MAXENT, random forests, or artificial neural network models?

- 8. In Section 2.2 Spatial database, the authors failed to provide the specific details of the input data, such as the dates/years in acquiring them, pre-processing processes, spatial resolution, and accuracies. In particular, did the authors utilize the long-term multi-temporal spatial data given that the mass movement data are from 2005 to now? How to deal with the data with different spatial resolution? Are the mass movement points observed in 17 years ago still active? It should be better to remain only those frequently-happened landslide points.

- 9. In Line 235~238, the authors mentioned that: "Since the mass movement inventory was stored as points, a grid was considered as one with mass movement if it contained at least one event of mass movement. If the grid unit did not include any mass movements, it was considered as one without any mass movements". What about the grids with more than one event of mass movement? They should be given more attentions than those grids with only one event.

- 10. In Line 313~316, the authors mentioned that: "Regarding the hyperparameter settings of this study, we set the parameters ‘num_boost_round’ as 500 and the ‘max_depth’ as 10 for all mass movements. Besides, the parameters ‘learning_rate’ is set to 0.1 for debris flow and rockfall and 0.2 for landslide". However, the authors need to clearly explain the detailed determination procedures of these parameters.

- 11. It is difficult to evaluate the validity/reliability of those susceptibility maps because of the limited description of validation processes (How the authors judged the “high precision” of those results?).

- 12. The accuracy values for both training and testing samples should be presented. The current results of this study can hardly be reviewed because of those problems about data and methodology.

- 13. It is suggested that the national risk hotspot regions should be identified by using the hotspot analysis tool, which can provide meaningful and interesting results.

- 14. Can it be possible for a reader or researcher to determine the mass movement susceptibility of their own cities from those result maps?

- 15. What is the unit of the "Differences" in the Figures 7 to 9.

Round 2

Reviewer 2 Report

Thank you for incorporating my previous comments and suggestions.